# Endothelial cell-initiated extravasation of cancer cells visualized in zebrafish

Masamitsu Kanada[1,*], Jinyan Zhang[1], Libo Yan[1], Takashi Sakurai[2] and Susumu Terakawa[1,**]

[1] Medical Photonics Research Center, Hamamatsu University School of Medicine, Handayama, Higashi-ku, Hamamatsu, Japan
[2] Electronics-inspired Interdiciplinary Research Institute, Toyohashi University of Technology, Hibarigaoka, Tempaku, Toyohashi, Japan
[*] Current affiliation: Department of Pediatrics, Stanford University School of Medicine, Stanford, CA, USA
[**] Current affiliation: Department of Health Science, Tokoha University, Mizuochi, Aoi-ku, Shizuoka, Japan

## ABSTRACT

The extravasation of cancer cells, a key step for distant metastasis, is thought to be initiated by disruption of the endothelial barrier by malignant cancer cells. An endothelial covering-type extravasation of cancer cells in addition to conventional cancer cell invasion-type extravasation was dynamically visualized in a zebrafish hematogenous metastasis model. The inhibition of VEGF-signaling impaired the invasion-type extravasation via inhibition of cancer cell polarization and motility. Paradoxically, the anti-angiogenic treatment showed the promotion, rather than the inhibition, of the endothelial covering-type extravasation of cancer cells, with structural changes in the endothelial walls. These findings may be a set of clues to the full understanding of the metastatic process as well as the metastatic acceleration by anti-angiogenic reagents observed in preclinical studies.

Corresponding author
Susumu Terakawa,
terakawa@sz.tokoha-u.ac.jp

## INTRODUCTION

Metastasis is the primary factor associated with the death of cancer patients. There is no therapeutic agent available to prevent this pathological step (*Gupta & Massague, 2006*). Metastatic progression proceeds by multiple steps: first, the development of vasculature inside a primary nest of tumor, intravasation of tumor cells into the newly developed leaky vasculature, survival of the cells under the stress in the systemic circulation, extravasation of the cells from the circulation, and finally proliferation at a secondary site in a distant tissue (*Nguyen, Bos & Massague, 2009*). These steps have been verified by studies of cancer cells or endothelial cells under *in vitro* culture conditions, or by examining preparations of fixed tissue specimens. Although histological or biochemical techniques may provide important information, such information is only validated at a certain point of time and thus compromises the interpretation on the dynamic aspects of metastasis. One of the difficulties in observing the behavior of cancer cells *in vivo* in mice by conventional high-resolution imaging techniques is the low transparency of the

tissue. Advanced techniques for intravital observations, such as two-photon microscopies, imaging chamber recording, fiber-optic fluorescence microendoscopies, have gradually enabled the visualization of the dynamic environmental changes accompanying tumor development at a cellular level (*Flusberg et al., 2005*; *Beerling et al., 2011*; *Ritsma et al., 2012*). However, no study has so far clearly shown the whole process of metastasis in mammalian tumor models at the cellular level.

A novel imaging technique was developed to overcome these difficulties in observing the dynamic process of cancer cell metastasis *in vivo* by taking advantage of the high transparency of zebrafish (*Stoletov et al., 2007*; *Stoletov et al., 2010*; *Zhang et al., 2013*). The zebrafish is an ideal vertebrate model for imaging, not only because of its optical transparency but also because a comparison of the zebrafish genome with that of a human revealed a remarkable conservation in the sequence of genes associated with the cell cycle, tumor suppression, proto-oncogenes, angiogenic factors, and extracellular matrix proteins (*Berghmans et al., 2005*; *Zon & Peterson, 2005*; *Stoletov & Klemke, 2008*). Highly metastatic cancer cells are often trapped in the capillaries and efficiently extravasated in the zebrafish, and an overexpression of the pro-metastatic gene "Twist" in cancer cells dramatically promotes their intravascular migration and extravasation (*Stoletov et al., 2010*).

The present study extended the zebrafish hematogenous metastasis model, and thereby made it possible to study the extravasation of human cancer cells, especially after forming severe emboli in the arterioles of zebrafish. The results obtained using a long-time fluorescence time-lapse recording system demonstrate that human cancer cells extravasate according to the manner generally accepted as an active invasion of a cancer cells. An extraordinary event occurred: a mass of cancer cells underwent embolus formation and then also extravasated via a covering with a layer of endothelial cells even in the absence of active invasion of the cancer cells. An electron microscopic study of a mouse lung metastasis model revealed similar cancer cell extravasation many years ago (*Lapis, Paku & Liotta, 1988*). A dynamic observation method demonstrated that the covering by endothelial cells is the major event in cancer cell extravasation. Furthermore, the live observation system confirmed that VEGF was associated with this manner of extravasation. Paradoxically, the treatment with an anti-angiogenic inhibitor shows the promotion, rather than the prevention, of the endothelial covering-type extravasation.

## MATERIALS AND METHODS

### Cell lines

DsRed2 (referred to as RFP) expressing HeLa cells (obtained from Anticancer) were cultured in RPMI-1640 supplemented with 10% FBS, 2 mM $l$-glutamine (Invitrogen), 1% Penicillin-Streptomycin (Invitrogen). The cells were incubated at 37 °C in 5% $CO_2$ in a humidified incubator.

### Zebrafish hematogenous metastasis model

Maintenance of the transgenic zebrafish and the experimental design for this study were approved by the Animal Welfare Committee of the Hamamatsu University School of

Medicine animal welfare. Zebrafish were maintained according to standard methods (*Westerfield, 1993*). The transgenic strain of zebrafish expressing enhanced green fluorescent protein (EGFP) under the *flk1 (VEGFR2)* promoter (*flk1: EGFP*) was obtained from the Zebrafish International Resource Center (*Jin et al., 2005*). Human cancer cells were microinjected into the zebrafish larvae, following the protocol reported previously with some modifications (*Stoletov et al., 2010*). Fish larvae were dechorionated and anesthetized with 0.006% tricaine 48 h post-fertilization (hpf; Sigma). Anesthetized larvae were then transferred onto an agarose gel for microinjection. RFP expressing cancer cells were detached from culture dishes using enzyme-free and PBS-based Cell Dissociation Buffer (13151-014, Gibco) and then were washed twice with PBS. Cancer cells were injected into the common cardinal vein using a tapered borosilicate glass capillary (1.0 mm in diameter, World Precision Instruments, Inc.) with a tip diameter of 20–40 μm (i.d.) connected to a 50 ml glass syringe. The position of the capillary tip was controlled by a manipulator (M-152, Narishige Scientific). The injected fish larvae were kept at 32 °C in the presence of 25 μg/ml dexamethasone (Sigma) for 5–7 h for immunosuppression only before observation because the presence of both tricaine and dexamethasone suppressed the heart beat of larvae. The larvae that formed severe emboli in the arterioles were selected under a fluorescence stereomicroscope (SZX16, Olympus), and used for further experiments.

## Live imaging of embolus-forming cancer cells and endothelial cells

A small drop of water containing an anesthetized larva was placed in a glass-bottom dish (36 mm, Matsunami, Gifu). One ml of low temperature melting agarose (E-3126-25, BM Equipment, Tokyo) containing 0.006% tricaine was added to the dish to hold the larva in a gel attached to the glass at the bottom. The dish was filled with 2 ml water containing 0.006% tricaine. The long time dual color time-lapse recording was carried out by using a homemade imaging system driven by ImageJ software (NIH, USA) or a microscope system in an incubator (BioStation, Nikon, Tokyo). The filter wheels (FW102; Tholabs, NJ) were controlled by the ImageJ software (Research Services Branch, National Institute of Mental Health, Bethesda, Maryland, USA; http://imagej.nih.gov/) through an IJSerial plugin (http://www.eslide.net/ijstage.php) for the dual color fluorescence imaging, and time-lapse images were captured using a QuickTime Capture plugin and Time-Lapse Video macros with some modifications (http://rsbweb.nih.gov/ij/plugins/qt-capture.html). The analog monochrome camera (WAT-120N+; Watec, Tokyo) was connected to the digital converter (ADVC-300; Canopus) and a PC via IEEE1394 interface. The background noise was decreased using an image processor (ARGUS-20; Hamamatsu Photonics, Hamamatsu). The temperature on the stage was maintained at 32 °C using a transparent heating plate (Kitazato, Fuji). In a standard recording, fluorescence images were captured every 5 min for a total time of up to 11 h. The spatial evaluation of cancer cells and endothelial cells was carried out using a confocal microscope (FV1000; Olympus). 3D stack images were taken by confocal microscopy (FV1000; Olympus) and processed using image processing package Fiji (http://fiji.sc/Fiji) (*Schindelin et al., 2012*).

## RNA interference, RT-qPCR

The siRNA sequence targeting human VEGF-A (referred here as VEGF) was the same as that designed in a previous report (*Takei et al., 2004*): 5′-GGA GUACCCUGAUGAGAUCdTdT-3′ (sense), 5′-GAUCUCAUCAGGGUACUCCdTdT-3′ (antisense). MISSION siRNA Universal Negative Control (Sigma) was used as a nonsense control siRNA. Relative mRNA amounts were quantified using iQ$^{TM}$ SYBR$^®$ Green Supermix (Bio-Rad Laboratories, Inc). Forward 5′-CCTGGTGGACATCTTCCAG GAGTA-3′ and reverse 5′-CTTGGTGAGGTTTGATCCGCATAA-3′ primers were used to detect VEGF, and forward 5′-AACGGATTTGGTCGTATTGGGC-3′ and reverse 5′-TTCTCAGCCTTGACGGTGCCAT-3′ primers were used to detect glyceraldehyde-3-phosphate dehydrogenase (GAPDH) mRNA. For comparative, quantitative analysis, transcript levels were normalized to the level of GAPDH and changes were determined. The comparative quantitation method ($\Delta\Delta Ct$) was used to compare the different samples and transformed to absolute values with $2^{\wedge}(-\Delta\Delta Ct)$ for obtaining relative fold changes. All assays were performed in triplicates. VEGF siRNA-treated RFP-HeLa cells were injected into the blood vessels 17–19 h after transfection. The cancer cell-injected larvae were kept at 32 °C in the presence of 25 µg/ml dexamethasone for 5–7 h for immunosuppression, and these cells were then observed for 11 h within 37 h after the transfection, so that VEGF was efficiently depleted during the observation period.

## Anti-VEGF treatment

Sunitinib (Toronto Research Chemicals, Toronto), an orally active VEGFR tyrosine kinase inhibitor, was dissolved in dimethyl sulfoxide (DMSO) to make a stock solution of 10 mM. The stock solution was diluted in water to attain a concentration of 5 µM. Sunitinib treatment was started immediately after cancer cell injection. The injected fish larvae were kept at 32 °C in the presence of 25 µg/ml dexamethasone (Sigma) and 5 µM sunitinib for 5–7 h for immunosuppression before observation. The cancer cell-injected larvae were then held in glass-bottom dishes with low temperature melting agarose on which water containing 0.006% tricaine and 5 µM sunitinib was added.

## *In vitro* observation of live cells

The cells were seeded in polymer-bottom dishes (Bio Medical Science, Japan), and time-lapse images were captured every minute for 12 h using BioStation (Nikon).

## Immunofluorescence

The cells in the polymer-bottom dishes were fixed (4% paraformaldehyde for 15 min), permeabilized (1% Triton X-100 for 15 min) and blocked (PBS containing 2% bovine serum albumin (Sigma, St Louis) for 30 min) for non specific immunostaining. The cells were then incubated with anti-vinculin mouse monoclonal antibody (ab18058, 1:200, Abcam, City) overnight at 4 °C, subsequently with anti-mouse-IgG conjugated with Alexa 488 (Invitrogen), and finally with 1 µg/ml Hoechst 33342 (Dojindo, Kamimashiki). The immunostained cells were observed under a confocal microscope (FV1000; Olympus, Hachioji).

## Chemotactic cell migration assay

The cells were seeded in BD Falcon™ FluoroBlok™ 24-Multiwell Insert Systems (351157; BD, NJ). The bottom wells were filled with chemoattractant-rich NIH3T3 conditioned medium. The cells were stained with calcein-AM (Dojindo) after 24-h incubation, and the number of stained cells in the bottom wells was counted in fluorescence images captured using an inverted fluorescence microscope.

## Electron microscopy

Fish larvae were fixed for scanning electron microscopy (SEM) by perfusion with 2% paraformaldehyde and 2.5% glutaraldehyde in PBS using the microinjection system for approximately 5 min, and then transferred into the same fixation buffer. Samples were dehydrated in a graded ethanol series, then frozen in liquid nitrogen and cracked into several pieces. These samples were then collected and freeze dried with $t$-butanol, and stained with $OsO_4$, and examined with a scanning electron microscope (S-4800; Hitachi, Tokyo).

# RESULTS

## Extravasation of embolus-forming human cancer cells

This study first examined the potential for the extravasation of human cervical cancer cells (HeLa). RFP-expressing HeLa (RFP-HeLa) cells were microinjected into the circulation, following the protocol previously reported (*Stoletov et al., 2010*). Unlike the previous study reporting that injected cancer cells are arrested in the thinner intersegmental vessels (*Stoletov et al., 2010*), RFP-HeLa cells formed severe emboli mostly in the thicker caudal artery immediately after injecting into the circulation. The clusters of cancer cells in the emboli extravasated and adhered to the tissue outside the blood vessels after 17–20 h (Fig. 1A). The spatial distribution of the adhering-cancer cells and blood vessel-forming endothelial cells was evaluated using a confocal microscope 1 day after the cancer cell injection to confirm that embolus-forming cancer cells actually extravasated and developed adhesion to the tissue outside the blood vessels. The optically sliced images clearly showed that endothelial tube structures were devoid of the mass of cancer cells, and all of the RFP-HeLa cells adhered to the tissue outside the blood vessels (Fig. 1B). Therefore, RFP-HeLa cells, which formed severe emboli in thicker caudal artery of zebrafish, have the ability to efficiently extravasate and adhere to the tissues surrounding the blood vessels.

## Two processes of extravasation in human cancer cells

The process of extravasation by the RFP-HeLa cells was observed using a long-time dual color time-lapse recording system to study mechanisms of extravasation of human cancer cells. The embolus-forming RFP-HeLa cells and endothelial cells could be observed for more than 11 h, which was long enough for these cells to exhibit slow behaviors. There were two distinctive processes of extravasation observed in RFP-HeLa cells (Fig. 2A). Some of the cancer cells actively invaded the vessel wall and penetrated the wall in

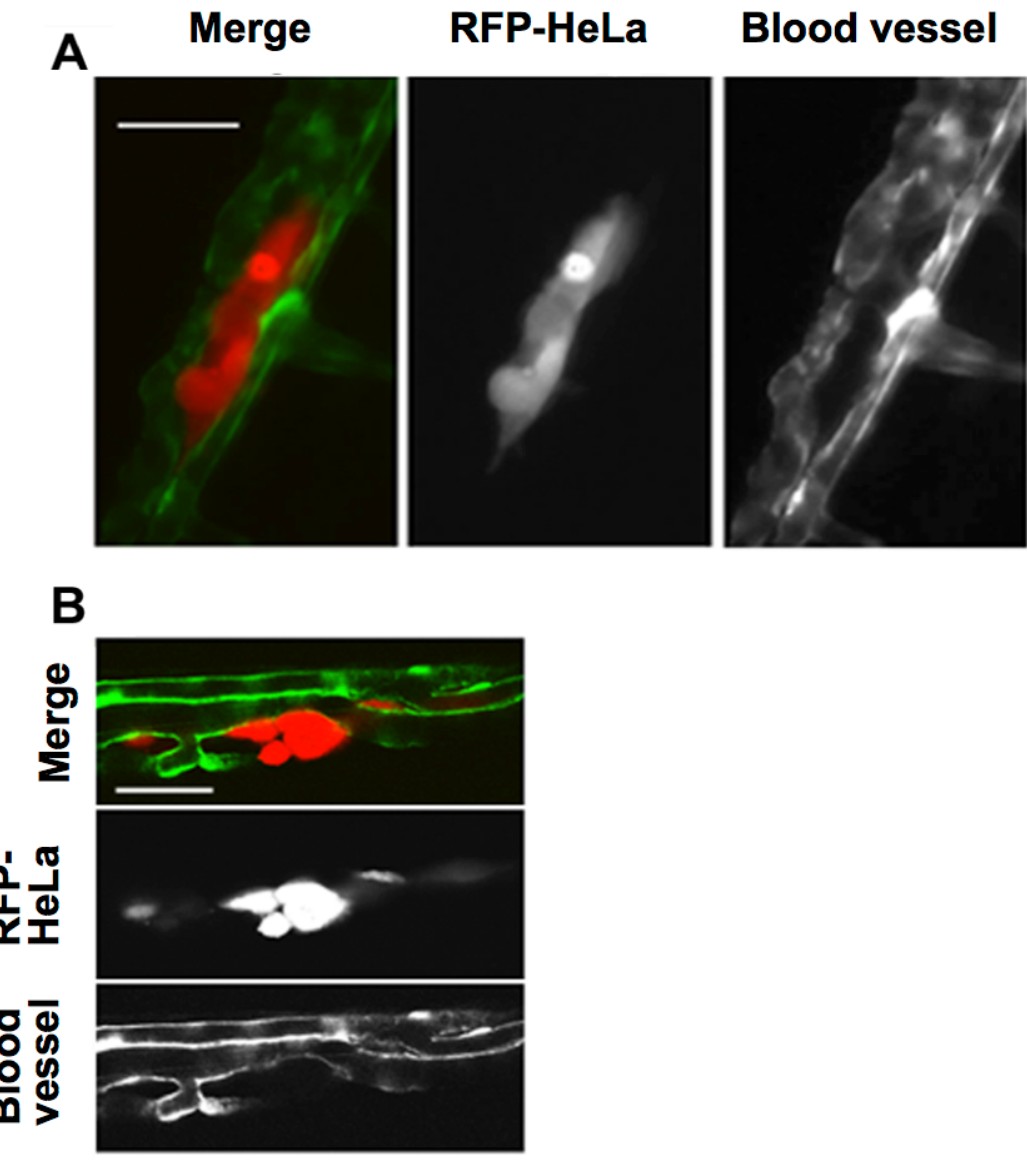

**Figure 1** **Extravasation of cancer cells after severe emboli formation in arterioles of zebrafish larvae.** (A) RFP-HeLa cells and endothelial cells in arterioles were examined under an epifluorescence microscope 17–20 h after formation of the emboli. (B) The spatial locations of the extravasated RFP-HeLa cells and blood vessels were examined under a confocal microscope. Bars, 100 μm (A) or 40 μm (B).

the process broadly accepted as the process of extravasation (*Stoletov et al., 2010*; *Weis et al., 2004*) (referred to as cancer cell invasion; Fig. 2A, left; Movie S1). In contrast, other masses of cancer cells seemed to be quiescent. These cells did not invade the endothelial cell layer. Instead, a new leaf of endothelial cells appeared and extended over the embolus-forming cells. The endothelial cells eventually covered the cancer cells on the vessel wall. Simultaneously, the original layer of endothelial cells gradually disappeared and the cluster of cancer cells spread to the tissue outside the blood vessels (referred to as endothelial covering; Fig. 2A, right; Movie S2). Intriguingly, 9 out of 17 extravasations were
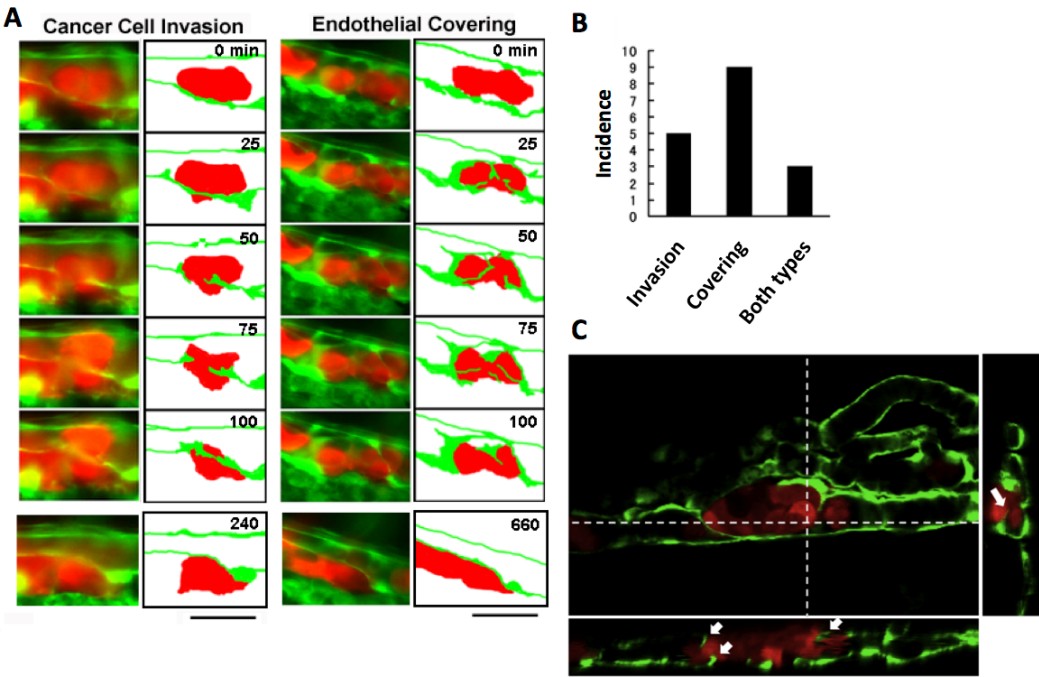

**Figure 2 Two representative processes of extravasation in cancer cells.** (A) Cancer cell invasion-type and endothelial covering-type extravasation in RFP-HeLa cells. Right panels show schematic outlines of the embolus-forming RFP-HeLa cells (red) and the surrounding endothelial cells (green). The numbers indicate the elapsed time in minutes. Bars, 40 μm. (B) The incidence of the 2 processes of extravasation. Seven larvae in which RFP-HeLa cells formed severe emboli were observed. Seventeen extravasation events were counted during the 11-h observations. (C) Side view of 3D reconstructed images. Some endothelial cells were extending over or penetrating through the embolus-forming RFP-HeLa cells (white arrows). Dashed lines in 2D image indicate locations of slicing along a vertical axis.

carried out in the absence of cancer cells' active invasion, while 5 were clearly accompanied by invasion of cancer cells. In the remaining 3 extravasations, embolus-forming cancer cells showed invasion and surrounding endothelial cells also newly extended over the cancer cells (Fig. 2B). The spatial distribution of embolus-forming cancer cells and newly spreading endothelial cells was further analyzed using 3D confocal microscopic images (47 slices, step size: 1 μm) that were taken at 10 h postadministration. Side views of the stack image and 3D reconstructed image clearly showed that some endothelial cells were extending over the cancer cells or penetrating into the cluster of cancer cells (Fig. 2C; Movie S3). These results suggest that efficient activation of endothelial cells by cancer cells is also important for extravasation, in addition to high motility of cancer cells, which has been regarded as one of the most crucial factors associated with malignancy.

Normal proliferating fibroblast 3T3 cells (NIH, USA) showed neither the invasion type nor the endothelial covering type extravasation in this model. Six clusters of these cells were in the vessels of 6 different individual fish for more than 11 h. All of them formed severe emboli similar to those of cancer cells, but showed neither the development of strong adhesions to the vascular walls nor the process of extravasation mentioned above. These cells were eventually pushed away by the blood flow. This suggests a difference to

exist in the expression of adhesion molecules between tumorigenic cells and normal cells, and adhesion to the endothelial walls is a key to induce extravasation for the circulating cancer cells after forming an embolus in the blood vessels.

## Effects of VEGF depletion on cancer cell properties and extravasation

Malignant tumors actively induce new blood vessels from surrounding tissues by secreting vascular endothelial growth factor (VEGF) to receive nutrients and oxygen. In addition, cancer cells may intravasate through this newly-formed leaky vasculature, leading to the metastasis in remote places (*Senger et al., 1993*; *Esser et al., 1998*). Hence we examined the effects of depletion of VEGF expressed in RFP-HeLa cells to study the role of VEGF in tumor cell extravasation. The cells were transfected with siRNA prior to their injection into the vessel of the fish. The efficient depletion of VEGF was confirmed by real-time quantitative PCR (RT-qPCR). All the 3 family members of VEGFA, which are secreted by cancer cells, have the same target sequence so that all the VEGFA (referred to as VEGF, here) were depleted in the cancer cells. Expression of VEGF was depleted to $29 \pm 6\%$ in comparison to the control obtained by using nonsense siRNA-treated cells (Fig. 3A). The effect of VEGF depletion on the cancer cell itself was then evaluated morphologically *in vitro*. The VEGF-depleted cells tended to aggregate with their motility decreased uniformly. They were flattened and their adhering surface became very large (Fig. 3B; Movie S5). Quantitative evaluation indicated that cell motility was reduced to $36.0 \pm 7.7\%$ in comparison to the control obtained by using nonsense siRNA-treated cells as reported previously (*Bachelder et al., 2003*) (Fig. 3C).

Vinculin, a focal adhesion protein, was immunostained in nonsense siRNA-treated and VEGF siRNA-treated RFP-HeLa cells to examine the effects of VEGF depletion on the adhesive property of cancer cells. Focal adhesions were uniformly formed at the cell periphery in VEGF depleted-cells, but only locally in control siRNA-treated cells so as to form a highly polarized shape (Fig. 3D). These results suggest that VEGF is involved not only in the activation of endothelial cells, but also in the migration of cancer cells via cell polarization.

We next examined the effects of VEGF depletion on extravasation *in vivo*. Like the normal RFP-HeLa cells, the VEGF-depleted cells immediately formed emboli after being injected into blood vessels, and then the endothelial cells migrated over the embolus-forming cancer cells, although the process was markedly delayed (Fig. 3E). Strikingly, the process of cancer cell invasion through the endothelial cells was completely suppressed during the 11-h observation period (Figs. 3E and 3F; Movie S6). Despite the severe inhibition of cancer cell invasion, six extravasation events were still observed in VEGF-depleted cells (Fig. 3F), thus suggesting that the clinically adopted anti-angiogenic strategy of VEGF targeting is insufficient for the prevention of metastasis. However, since RNA interference does not completely suppress target molecule expression compared to that of gene knockout, we cannot rule out the possibility that a residual amount of VEGF could be enough to activate surrounding endothelial cells.

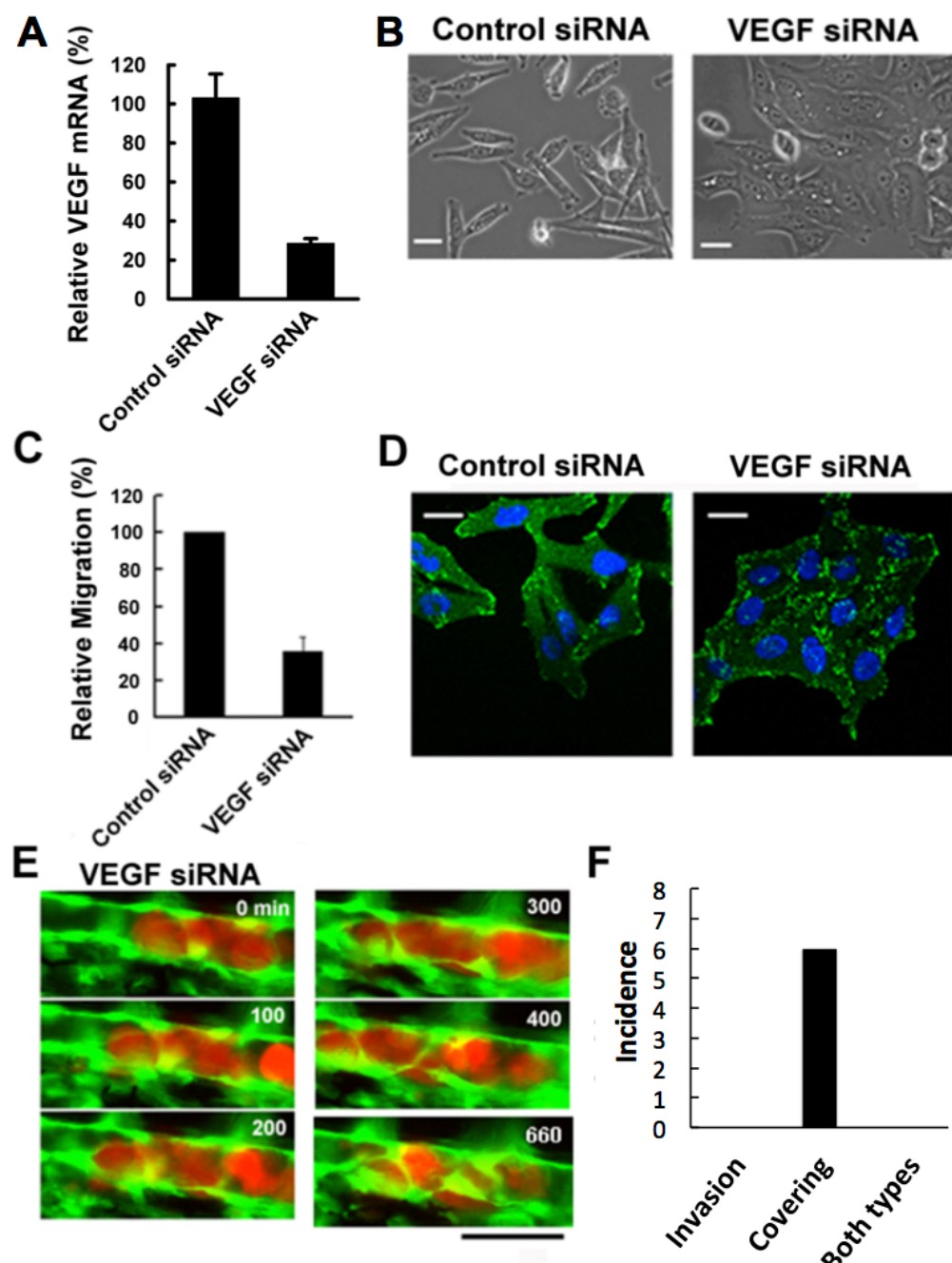

**Figure 3 Effects of VEGF depletion on cancer cell properties and extravasation.** (A) The depletion of VEGF in RFP-HeLa cells was confirmed by RT-qPCR 24 h after the transfection with siRNA. GAPDH was amplified as an internal control. (B) Phase contrast images of control siRNA-treated and VEGF-depleted RFP-HeLa cells in culture. (C) Quantitative evaluation of chemotactic migration in control siRNA-treated and VEGF-depleted RFP-HeLa cells (mean ± SD, $n = 3$). (D) Confocal microscopic images of control siRNA-treated and VEGF-depleted RFP-HeLa cells showing the distribution of vinculin (green) and DNA (blue). (E) The extravasation of VEGF-depleted RFP-HeLa cells. The numbers indicate the elapsed time in minutes. (F) Incidence of the 2 processes of extravasation. Nine larvae in which VEGF-depleted RFP-HeLa cells formed severe emboli were observed. Six extravasation events were counted during 11-h observations. Bars, 20 μm (B, D) or 40 μm (E).

## Effects of a multi-targeted anti-angiogenic kinase inhibitor on cancer cell properties and extravasation

VEGF depletion in RFP-HeLa cells only delayed the endothelial covering-type extravasation, while completely inhibiting the cancer cell invasion-type extravasation. Therefore, we anticipated that a potent inhibitor of tumor angiogenesis via VEGF signaling could completely inhibit both types of extravasation. Sunitinib (Sutent®; Pfizer Inc.), which effectively suppresses angiogenesis by inhibiting signaling from the receptors for VEGF and for platelet derived growth factor (PDGF) (*Bergers et al., 2003*; *Pietras & Hanahan, 2005*), was orally administered to the zebrafish larvae. The thinner intersegmental vessels were severely deteriorated by the overnight treatment with sunitinib (Fig. 4A). However, the thicker blood vessels were not affected morphologically, suggesting that the effect of sunitinib is limited to the newly-formed vulnerable vessels. Endothelial cells in the mature vasculature are independent of VEGF signaling for survival (*Gerber et al., 1999*; *Lee et al., 2007*). Sunitinib inhibits the VEGF receptor, so the treatment of cells with sunitinib should affect the cancer cells like VEGF depletion. As expected, the cells examined *in vitro* adhered to an obviously larger area of the substrate, became less motile, and tended to aggregate in the presence of sunitinib, as the VEGF-depleted cells did (Fig. S1A). In addition, the sunitinib treatment reduced the cell motility by $35.0 \pm 5.5\%$ (Fig. S1B). Focal adhesions were uniformly formed at the cell periphery in sunitinib-treated cells, as in VEGF-depleted cells (Fig. S1C).

The effect of sunitinib on extravasation was examined *in vivo*. Sunitinib treatment completely suppressed the process of cancer cell invasion-type extravasation as expected. In an unexpected development, the incidence of endothelial covering-type extravasation of RFP-HeLa cells was not affected by treatment with sunitinib, although the process was markedly delayed, similarly to the extravasation of VEGF-depleted cells (Fig. 4B; Movie S7). The area of cancer cells that were covered by the endothelial cells in the presence of sunitinib was markedly larger than that in normal or VEGF-depleted RFP-HeLa cells in most events of extravasation, and the blood vessel walls concurrently, but slowly, moved toward the embolus-forming RFP-HeLa cells (Fig. 4B; Movie S7). Therefore, the sum of the area of extravasated cancer cells was calculated from 7 movies that were recorded in RFP-HeLa cells with and without sunitinib treatments to quantitatively compare the whole volume of cancer cells extravasated (Fig. S1D). Although the incidence of endothelial covering-type extravasation in the presence of sunitinib (9 incidences in 7 larvae) was the same as that observed without treatment of sunitinib (Figs. 2B and 4C), sunitinib treatment increased the total volume of extravasated cancer cells to 153% of the volume of extravasated cells without sunitinib treatment during 11-h observation period (Fig. 4D). These results suggest that the anti-angiogenic inhibitor sunitinib can completely suppress the cancer cell invasion-type extravasation without having strong effects on the incidence of endothelial covering-type extravasation. Paradoxically, the total volume of the extravasated cancer cells increased in the presence of sunitinib, thus suggesting that sunitinib accelerates the process independently of the VEGF signaling. Moreover, the endothelial cells apparently activated by sunitinib were studied morphologically using a scanning electron microscope. Many

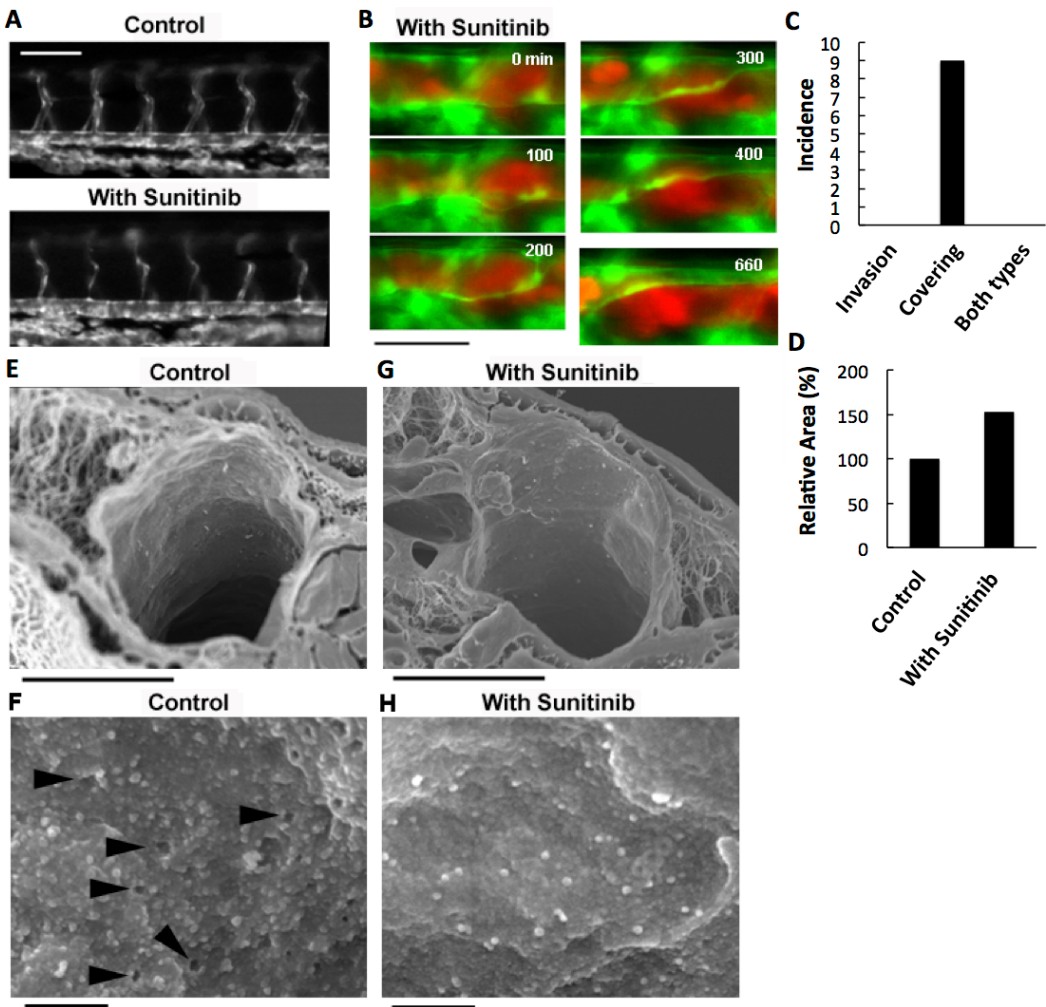

**Figure 4 Effects of anti-angiogenic inhibitor, sunitinib, on the vasculature of zebrafish larvae and cancer cell extravasation.** (A) Intersegmental vessels were severely deteriorated by treatment with sunitinib. (B) Extravasation of RFP-HeLa cells in the presence of sunitinib. The numbers indicate the elapsed time in minutes. (C) The incidence of the 2 processes of extravasation. Seven larvae in which RFP-HeLa cells formed severe emboli were observed in the presence of sunitinib. Nine extravasation events were counted during 11-h observations. (D) The sum of the area of extravasated cancer cell images was calculated from all of the 7 movies recorded in RFP-HeLa cells with and without sunitinib treatments. Bars, 100 μm (A) or 40 μm (B). (E, G) Scanning electron micrographs of the arteries in the control and sunitinib-treated larvae. Sunitinib-treated larvae showed thicker vascular walls. (F) The magnified image of the luminal face on the endothelial wall of a control larva. The endothelial wall showed many holes or fenestration-like structures (arrowhead). (H) The magnified image of the luminal face on the endothelial wall of a sunitinib-treated larva. The endothelial wall showed no fenestration-like structures. Bars, 10 μm (E, G) or 0.5 μm (F, H).

protrusions were observed on the luminal face of endothelial walls of arteries in zebrafish, and interestingly, many small holes were found in some regions on the endothelial wall (Fig. 4F). These holes are reminiscent of fenestrated endothelial walls in tumor vasculature or normal microvasculature of mice (*Inai et al., 2004*; *Kamba et al., 2006*;

*Mazzone et al., 2009*). Sunitinib-treated zebrafish larvae showed no hole on the endothelial walls of arteries with fewer protrusions (Fig. 4H), and also showed prominently thicker endothelial walls (Fig. 4G).

## DISCUSSION

Most reports addressing tumor metastasis present evidence supporting the assumption that extravasation and intravasation are initiated by disruption of the endothelial barrier by malignant cancer cells via VEGF secretion and invasion (*Weis et al., 2004*). The present study observed a new process, potentially revising the conventional understanding of the mechanism of metastasis. A unique hematogenous metastasis model in the transparent zebrafish was used to observe the extravasation of human cancer cells after embolus formation at a cellular level. This approach revealed that extravasation could be provoked by a mass of cancer cells, in addition to the conventional manner in which a single cell individually extravasates like the invasion of immune cells (*Middleton et al., 2002*). The results confirmed that the extravasation is initiated by extension of a new leaf of endothelial cells covering the mass of tumor cells in the capillary (*Lapis, Paku & Liotta, 1988*). The invasion of cancer cells through the endothelium occurred as in the widely accepted model of cancer metastasis. The study demonstrated that VEGF produced by the tumor cells is involved in regulation of their migration by affecting the cell adhesion and polarization, and VEGF-depletion completely suppressed the cancer cell invasion-type extravasation. On the other hand, the endothelial covering-type extravasation was not inhibited but simply delayed by VEGF depletion, suggesting that the process is partially dependent on VEGF and redundant pathways could compensate to complete the extravasation. Sunitinib, an anti-angiogenic inhibitor, suppresses the migration and proliferation of endothelial cells and the migration of the cancer cells simultaneously during the extravasation. Surprisingly, sunitinib treatment had no significant effect on the process of endothelial covering in the current metastasis model system, although it obviously inhibited the new formation of intersegmental vasculature in the zebrafish. Furthermore, ultrastructural observations of the luminal face of an artery in the zebrafish showed that the sunitinib treatment eliminated the fenestration-like holes on the endothelium and induced thicker vascular walls. These findings suggest that sunitinib induces vascular maturation independent of VEGF, while deteriorating newly-formed vasculature. VEGF-independent vascular remodeling could be a key regulatory mechanism underlying the extravasation of cancer cells. Since sunitinib is a multi targeted kinase inhibitor, this vascular remodeling could be its effects on PDGFR or other targets. To address this possibility, VEGF specific inhibition such as anti-VEGF antibody would be ideal, although administrating adequate amount of antibody into the vasculature of small zebrafish larvae is technically difficult.

The current findings also suggest that cancer cells have the potential to extravasate not only in capillary vessels but also in mature blood vessels by forming clusters which induce vascular remodeling. It is worth noting that the extravasation of cancer cells as clusters is likely to increase the chance of cancer cells to survive in the tissue outside the vasculature via secretion of trophic factors for their growth or signal molecules for the

immune tolerance in comparison to the extravasation of a single cell. For these reasons, the capability of cancer cells to activate endothelial cells has to be taken into consideration as an aspect of malignancy, in addition to cancer cell invasiveness.

Surprisingly, a VEGF-targeted inhibitor promoted the process of endothelial covering over embolic cancer cells independently of VEGF, rather than inhibiting it. Similar evidence has been presented in some preclinical studies, showing that VEGF-targeted inhibition promotes tumor invasiveness and metastasis (*Ebos et al., 2009*; *Pàez-Ribes et al., 2009*). The explanation for these apparently paradoxical effects observed in these studies is still controversial (*Loges et al., 2009*). *Pàez-Ribes et al. (2009)* demonstrated that it appears to be an adaptive/evasive response by the tumor cells triggered by a disruption of the tumor vasculature. One plausible mechanism to trigger the adaptation of the tumor cells is tumor hypoxia (*Brahimi-Horn, Chiche & Pouyssegur, 2007*). The anti-angiogenic treatment disrupts tumor vasculature in the initial phase, but a mechanism of evasive resistance to the anti-angiogenic treatment is then switched on to enable revascularization via alternative pro-angiogenic signals that increase local invasiveness and/or enhance distant metastasis. Hypoxia is an effective driving force for the evasive resistance of tumors through stabilization of hypoxia inducible factor-1 (HIF-1) (*Semenza, 2003*). Another study also reported that sunitinib, a VEGF-targeted inhibitor, promotes tumor metastasis in a preclinical model. Immunocompromised mice were pretreated or treated with sunitinib immediately after intravenous inoculation with tumor cells (*Ebos et al., 2009*). Intriguingly, the anti-angiogenic treatment increased the formation of metastatic foci, and shortened the overall survival time of the mice. This acceleration of metastasis cannot be explained solely by the mechanism of aforementioned hypoxia-related effects on primary tumors. Therefore, the anti-angiogenic VEGF-targeted inhibition may change the nature of of the vasculature and increasing the probability of cancer cell lodging and extravasation. The VEGF-disruption in mice can lead to a vessel disintegration, and render the endothelium prothrombotic (*Lee et al., 2007*), which probably increases the number of places where cancer cells lodge.

Many investigations reveal that there are multiple types of endothelial cells that have distinct molecular signatures (*Mazzone et al., 2009*; *Phng & Gerhardt, 2009*). Notch pathway is implicated in the regulation of VEGF pathway, and its up-regulation was suggested to contribute to VEGF-independent angiogenesis. This reciprocal regulation of the endothelial cell types is explained using the tip/stalk model (*Noguera-Troise et al., 2006*; *Ridgway et al., 2006*; *Scehnet et al., 2007*). In addition, *Mazzone et al. (2009)* reported that a heterozygous deficiency of the oxygen-sensing prolyl hydroxylase domain protein2 (PHD2) reverts the abnormal tumor vasculature formed in the tumor-burdened mice to the mature and stable one, containing an orderly formed tight monolayer of endothelial cell known as "phalanx cells." The sunitinib-treated endothelial cells in our model are reminiscent of the "phalanx cells" in the vascular walls. Based on the theory regarding endothelial cell types and resistance to anti-angiogenic therapy, the current findings provide a model to explain the VEGF-independent extravasation of cancer cells. The VEGF dependency and fenestration reveals that 2 types of endothelial cell populations

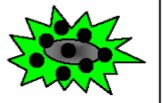

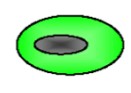

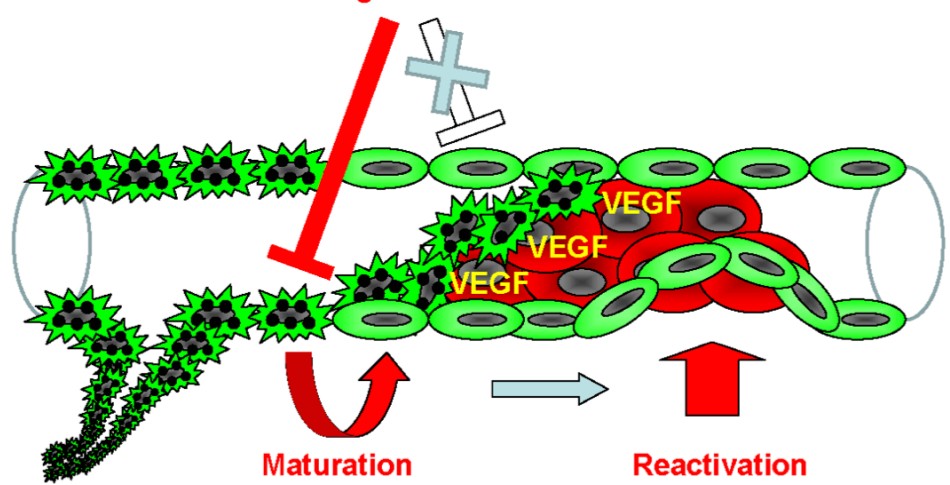

**Figure 5 Model of the endothelial coverage regulation via vascular homeostasis maintenance.** Active EC in the normal vasculature, which is VEGF-dependent and fenestrated, and quiescent EC, which is VEGF-independent and unfenestrated, coexist in the same blood vessel. VEGF-targeted inhibitors only affect the active ECs, thus resulting in deterioration of new vasculature via apoptosis or vascular maturation. In contrast, the quiescent ECs that contain mature vasculature are reactivated independently of VEGF in order to maintain vascular homeostasis.

are present in the vasculature: one is the VEGF dependent and fenestrated-type, which are defined as active endothelial cells that form new vasculature, and the other is the VEGF independent and unfenestrated-type, which are quiescent endothelial cells that form a mature vasculature. The application of anti-angiogenic inhibitors causes the new vasculature to deteriorate by apoptosis of the active endothelial cells or by vascular maturation. In turn, the quiescent endothelial cells could be reactivated for reconstruction of vessels independently of VEGF in order to maintain the homeostasis of the vasculature (Fig. 5). Some clinical or preclinical studies show that pro-angiogenic factors other than VEGF are induced by the anti-angiogenic treatment (*Willett et al., 2005*; *Ebos et al., 2007*; *Casanovas et al., 2005*). This evidence supports the current model that a disruption of neovascular endothelium by anti-angiogenic drugs promotes endothelial covering-type extravasation via reactivation of quiescent endothelial cells.

Anti-VEGF drugs such as the monoclonal anti-VEGF antibody bevacizumab (*Hurwitz et al., 2004*; *Miller et al., 2007*) and the multi-targeted receptor tyrosine kinase inhibitors sunitinib (*Demetri et al., 2006*; *Motzer et al., 2006*) and sorafenib (*Abou-Alfa et al., 2006*; *Escudier, 2007*; *Grepin & Pages, 2010*) prolong the life of some cancer patients, but the clinical benefits of the treatment are relatively modest and usually prolong the overall

survival of cancer patients only by a matter of months without offering an enduring cure (*Grepin & Pages, 2010*; *Kerbel, 2008*), and in some cases it may shorten the survival by facilitating the tumor invasiveness and metastasis. Although the mechanisms of the resistance to anti-angiogenic treatments and acceleration of metastasis are still under investigation, intrinsic tumor resistance or acquired resistance are proposed as possible mechanisms.

It is important to understand the complicated mechanism of vascular homeostasis during the anti-angiogenic treatment of such tumors, because the effects of anti-angiogenic drugs on the response of endothelial cells and on the intravasation and extravasation play a key role in each step of metastasis.

## ACKNOWLEDGEMENTS

We would like to thank Ms. Yoko Kumakiri for her excellent technical assistance on the electron microscopic study, Dr. Yuichi Hiratsuka for useful suggestions on ImageJ, Dr. Kaoru Kato for useful suggestions on microinjection, and The Zebrafish International Resource Center for providing the transgenic zebrafish.

### Funding

This work was supported by the Japan Society for the Promotion of Science (JSPS) fellowship to MK, a grant from the Japan Science and Technology Agency (JST) to ST and the intramural project "Cancer research by optical means." The Zebrafish International Resource Center is supported by NIH-NCRR (P40 RR012546) for publicly providing the transgenic zebrafish. The funders had no role in study design, data collection and analysis, decision to publish, or preparation of the manuscript.

### Grant Disclosures

The following grant information was disclosed by the authors:
The Japan Society for the Promotion of Science (JSPS).
The Japan Science and Technology Agency (JST).

### Competing Interests

The authors declare there are no competing interests.

### Author Contributions

- Masamitsu Kanada conceived and designed the experiments, performed the experiments, analyzed the data, contributed reagents/materials/analysis tools, wrote the paper, prepared figures and/or tables, reviewed drafts of the paper.
- Jinyan Zhang performed the experiments, analyzed the data, contributed reagents/materials/analysis tools, wrote the paper.
- Libo Yan performed the experiments, analyzed the data, contributed reagents/materials/analysis tools.

- Takashi Sakurai contributed reagents/materials/analysis tools, *in vivo* microscopy.
- Susumu Terakawa conceived and designed the experiments, contributed reagents/materials/analysis tools, wrote the paper, reviewed drafts of the paper.

## Animal Ethics

The following information was supplied relating to ethical approvals (i.e., approving body and any reference numbers):

Maintenance of the transgenic zebrafish and the experimental design for this study were approved by the Hamamatsu University School of Medicine animal welfare.

## Supplemental Information

Supplemental information for this article can be found online at http://dx.doi.org/10.7717/peerj.688#supplemental-information.

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
