# Peer review of "Endothelial cell-initiated extravasation of cancer cells visualized in zebrafish"

_PeerJ, doi:10.7717/peerj.688_

## Round 0.1 · original submission · Major Revisions

The study is a scientific argument on the conventionally well reported cancer cell extravasation, as the authors call "endothelial covering-type extravasation of cancer cells". Please address the comments of the reviewers in your revision.

·

Basic reporting

No comments

Experimental design

The experimental design is very good and properly described.

Validity of the findings

The finding are very well supported and the iconography is excellent!

Additional comments

I agree that the author have demonstrated that endothelia covering-type extravasation happens in absence of VEGF and therefore is likely to be one of the factors allowing tumour progression under anti VEGf treatment. I also agree that they have demonstrated that more cells extravasate. I am only doubtful whether this account for a more aggressive tumour as, after all, the invasion type is instead absent absent. Perhaps the author could briefly comment on this issue instead of concluding that could explain a more aggressive behaviour.

·

Basic reporting

no comments

Experimental design

no comments

Validity of the findings

no comments

Additional comments

The paper by Kanada et al., is a very well written paper addressing some important concerns/questions in the field of angiogenesis. The authors have shown that there are 2 types of extravasation using zebrafish models: endothelial covering type extravasation and invasion type extravasation. The authors have used siRNA approach and therapeutic approach to address the role of VEGF in metastasis. Overall the experiments are very convincing and the data supports the conclusions. Below are some of the concerns that needs to be addressed
1. The authors state that cell morphology and motility are dependent on VEGF involving intracellular signaling. Do we know how much VEGF is in the supernatant that has been added? Can we titrate different amounts of VEGF exogenously to know if it is for sure intracellular mediated?
2. With regards to the siRNA experiments the authors claim that inspite of severe inhibition of cancer cell invasion VEGF-depleted cells extravasated like normal within 60 hr. Since these are siRNA experiments, do we know how long the RNA knock down is maintained in these cells? The knock down could be transient and that is why you might see invasion at 60hr. In addition can authors use an anti-VEGF antibody to address this concern?
3. The authors also claim that extravasation is provoked by mass of cancer cells and not by single cells. There is not sufficient evidence in the paper to come to this conclusion. Have you tried using other cells or single suspension cells to confirm this? In addition how to NIH3T3 cells behave, do they form clumps too?
4. Sunitinib is sometimes referred as VEGF-targeted inhibitor; I would recommend addressing it as a multi targeted kinase inhibitor as there are many targets that Sunitinib hits. In addition the differences we see with Sunitinib could be its effects on PDGFRb and other targets it hits. This needs to be explicitly stated.
5. Anti-VEGF inhibitor instead of Sunitinib to address the question of role of VEGF in metastasis would be ideal
6. The lines 266-281 talk about the concept of VEGF dependent and independent cells. The way that authors talk, it sounds like it is a novel observation. This has been reported several times and they are usually described as tip and stalk cells. This has to be clarified.

---

## Round 0.2 · accepted · Accept

The Zebrafish materials the authors have made and evidenced should be avaliable for the request in future,as they have deposited in the HU.